# Sensitivity to gait improvement after levodopa intake in Parkinson's disease: A comparison study among synthetic kinematic indices

**Emahnuel Troisi Lopez**[1], **Roberta Minino**[1], **Pierpaolo Sorrentino**[2,3], **Valentino Manzo**[4], **Domenico Tafuri**[1], **Giuseppe Sorrentino**[1,3,5]*, **Marianna Liparoti**[1]

**1** Department of Motor Sciences and Wellness, University of Naples "Parthenope", Naples, Italy, **2** Institut de Neuroscience des Systemès, Aix-Marseille University, Marseille, France, **3** Institute of Applied Sciences and Intelligent Systems, CNR, Pozzuoli (NA), Italy, **4** Alzheimer Unit and Movement Disorders Clinic, Department of Neurology, Cardarelli Hospital, Naples, Italy, **5** Institute for Diagnosis and Care, Hermitage Capodimonte, Naples, Italy

☯ These authors contributed equally to this work.
* giuseppe.sorrentino@uniparthenope.it

**Data Availability Statement:** All relevant data are within the paper and its Supporting Information files.

## Abstract

The synthetic indices are widely used to describe balance and stability during gait. Some of these are employed to describe the gait features in Parkinson's disease (PD). However, the results are sometimes inconsistent, and the same indices are rarely used to compare the individuals affected by PD before and after levodopa intake (OFF and ON condition, respectively). Our aim was to investigate which synthetic measure among Harmonic Ratio, Jerk Ratio, Golden Ratio and Trunk Displacement Index is representative of gait stability and harmony, and which of these are more sensitive to the variations between OFF and ON condition. We found that all indices, except the Jerk Ratio, significantly improve after levodopa. Only the improvement of the Trunk Displacement Index showed a direct correlation with the motor improvement measured through the clinical scale UPDRS-III (Unified Parkinson's Disease Rating Scale–part III). In conclusion, we suggest that the synthetic indices can be useful to detect motor changes induced by, but not all of them clearly correlate with the clinical changes achieved with the levodopa administration. In our analysis, only the Trunk Displacement Index was able to show a clear relationship with the PD clinical motor improvement.

## Introduction

Parkinson's disease (PD) is an age-related neurodegenerative pathology, characterized by nigrostriatal dopaminergic degeneration [1]. Lack of dopamine causes motor system malfunctions such as bradykinesia, rigidity, tremor, gait disorders and postural instability with the consequent high risk of fall [2]. A temporary inhibition of symptoms occurs after taking levodopa (L-DOPA), which is currently the most effective symptomatic treatment [3, 4].

In order to rate the disease severity and to optimize the therapeutic strategy, it is crucial to have reliable and replicable scales for assessing the global clinical condition of the PD patients.

**Funding:** The authors received no specific funding for this work.

**Competing interests:** The authors have declared that no competing interests exist.

The Unified Parkinson's Disease Rating Scale (UPDRS) is the most commonly used [5]. The UPDRS consists of four parts, the third of which (UPDRS-III) specifically assesses motor impairment, and it can be employed before and after L-DOPA medication (OFF and ON condition, respectively), in order to assess the motor response to treatment. However, this approach is based on the subjective clinician's assessment of the motor state and may not truly represent the patients' motor impairment. Therefore, a less operator-dependent approach capable of providing a quantitative assessment of motor deficits is highly needed.

Gait analysis (GA) is a widely used methodology to study human locomotion. It is employed to analyse gait features in healthy people [5] and in individuals affected by both non-motor [6, 7] and motor diseases, including PD [3, 8–11]. Several technologies are exploited to gather data from gait; accelerometers, gyroscopes, magnetometers are commonly employed sensors in 3D analysis, but the gold standard for movement evaluation is represented by stereophotogrammetric systems [12]. In fact, through this approach it is possible to acquire spatio-temporal and kinematic parameters of high precision and reliability [13–15].

Recently, many studies have turned to analyse more synthetic measures in order to obtain an overall assessment of gait, based on features of gait like harmony and fluidity (or smoothness) [16, 17], commonly analysed through accelerometers, and occasionally using stereophotogrammetric systems. This approach has been applied especially in studying individuals affected by movement disorders, including PD [18–22]. The most commonly employed indices to asses harmony and smoothness of gait are: the Harmonic Ratio (HR) [23–25], the Jerk Ratio (JR) [26, 27] and the Golden Ratio (GR) [28, 29]. Very recently, we have implemented a new gait parameter, called Trunk Displacement Index (TDI) [21], that assesses the relationship between trunk and Centre of Mass (COM) oscillations.

The HR is based on harmonic theory and analyses the periodicity of acceleration signal and its definition is debated in gait analysis [30]. It is commonly described as a measure to quantify smoothness of walking [23, 31, 32] and quantify walking stability [18, 25, 33], but some author addresses its significance as a measure of symmetry between steps [30] or rhythmicity of the accelerations [25]. Higher values of HR are usually observed in young individuals when compared to elderly people [31, 34, 35]. With regard to individuals affected by PD, Castiglia et al., (2021) analysed HR in patients during ON phase. The authors reported higher anteroposterior HR values in PD with respect to healthy individuals matched for age and walking speed. Furthermore, they highlighted that anteroposterior HR could result as a useful marker to characterise falling risk [36]. Accordingly, further studies on PD individuals in ON condition showed lower HR values compared to healthy age-matched controls [25, 37].

The JR is a measure commonly used to assess smoothness of movement [38, 39] during gait, taking into consideration the jerk (the third derivative of position with respect to time) of the body through three-dimensional space. Similarly to HR, JR has been successfully exploited to distinguish between young and elderly individuals, with the former showing smoother movements (low JR values) [39]. Buckley et al. showed higher JR values in subjects with PD when compared to healthy controls, demonstrating the importance of upper body variables during gait, in conjunction to the spatiotemporal gait parameters [40]. However, the authors did not specify if the patients were tested in OFF or ON condition [26].

The GR, represented by the Greek letter *phi* (ϕ), was identified in human gait by Iosa et al. [29]. ϕ is a well-known mathematical proportion that describes a fractal harmonic structure [41, 42]. In particular, the authors hypothesized that the proportion between specific gait phases could comply with the value expressed by ϕ. Subsequently, the topic was explored further, and the authors hypothesized that the human anthropometric proportions have evolved in such a way to facilitate golden proportion in gait [43]. Furthermore, they hypothesised that the neural network comprising the cerebellum, globus pallidus spinal cord is what regulates

the harmonious golden ratio rhythm [44], and that this rhythm is altered in people with cerebellar ataxia [45] and people with Parkinson's disease [46]. In this regard, the authors, using a stereophotogrammetric system, performed a gait analysis in patients with PD and healthy controls, confirming their hypothesis and highlighting the presence of harmonic properties in human walking. They also demonstrated that harmonic proportions of gait were reduced in PD patients in both OFF and ON condition, compared to healthy controls [46].

Finally, the TDI, a recently developed measure that we introduced in a previous study, is an adimensional index able to quantify the displacement of the trunk in relation to the COM [21]. Higher TDI values indicate wide trunk displacement with respect to the COM trajectory, expressing low postural control. Its conception originated from the idea that through the evolution and the transition from quadrupedal to bipedal locomotion the positions of the trunk and the COM [47, 48] changed and the total weight of the body moved on two limbs, increasing the complexity of the task of keeping balance. Intuitively, it is expected that the trunk oscillation should not be too wide compared to the COM movement. In our previous study we showed how the TDI could distinguish the PD individuals before and after a sub-clinical dose of L-DOPA intake, with the PD patients in OFF condition exhibiting higher TDI values when compared to PD patients in ON condition [18].

As shown, synthetic measures have often been used to assess PD gait. However, these studies were carried out regardless of the ON or OFF condition and the differences between the two states have been poorly investigated. The aim of our work was to compare the sensitivity of the aforementioned measures to clinical motor changes. In particular, through a stereophotogrammetric system, we calculated the values of these synthetic measures, in individuals affected by PD, before and after L-DOPA intake. Furthermore, in order to assess their clinical meaning, we tested whether there was a relationship between the indices analysed and the UPDRS-III clinical scale scores.

## Materials and methods

### Subjects

We recruited twenty-one patients (Table 1) affected by PD (diagnosis defined according to the United Kingdom Parkinson's Disease Brain Bank criteria) [49]. Patients were acquired from July 22, 2020 to August 10, 2020. The following inclusion criteria were considered: i)

**Table 1. Demographic, neuropsychological, and clinical characteristics of the patients affected by Parkinson's disease (PD).**

| Demographic data | PD | | p-value |
|---|---|---|---|
| Age (years) | 64.4 (± 11.6) | | - |
| Gender (m/f ratio) | 16/5 | | - |
| Neuropsychological data | | | |
| MMSE | 28.1 (± 1.6) | | - |
| FAB | 16.3 (± 1.8) | | - |
| BDI | 6 (± 4.4) | | - |
| Clinical data | PDoff | PDon | |
| UPDRS-III | 28.5 (± 16) | 16 (± 9.3) | < 0.001 |
| Disease duration (months) | 86.4 (± 47.3) | | - |

Clinical assessment was compared within the group before (PDoff) and after (PDon) L-DOPA administration. Mini mental state examination (MMSE), frontal assessment battery (FAB), Beck's depression inventory (BDI), unified Parkinson's disease rating scale part III (UPDRS-III). Value expressed as mean (± standard deviation).

minimum age of 45 years or older; ii) Hoehn & Yahr (H&Y) score $\leq$ 3 in "OFF" state; iii) disease duration < 10 years; iv) presence of antiparkinsonian treatment at a stable dosage. Exclusion criteria included: i) Mini-Mental State Examination (MMSE) score < 24 [50]; ii) Frontal Assessment Battery (FAB) score < 12 [51]; iii) Beck Depression Inventory II (BDI-II) > 13 [52]; iv) presence of additional neurological or psychiatric disorders; v) assumption of additional psychoactive drugs; vi) any other physical or medical conditions causing walking impairment.

According to the declaration of Helsinki, an informed consent was obtained from all participants. The study was approved by the AORN "A. Cardarelli" Ethic Committee (protocol number: 00019628) on July 21, 2020.

## Intervention

The protocol required to record each subject two times, before (OFF state) and after (ON state) L-DOPA intake. Specifically, the patients in OFF state did not assume L-DOPA in the last 14–16 hours (PDoff group), while the second recording was performed on the same individuals who assumed a subclinical dose (defined as half of their usual morning intake) of L-DOPA (Melevodopa + Carbidopa) (PDon group) 40 minutes before the acquisition. Each acquisition was preceded by an UPDRS-III test. Specifically, patients were instructed to walk forth and back continuously at self-selected speed, through a measured space of 10 meters (Fig 1, left panel). This made it possible to record at least six trials for each subject and condition. This number of recordings represents a good compromise to avoid participants fatigue [53] during the execution of the task and to obtain information on the kinematic of the gait. Precisely, participants were asked to start the walking, without knowing when they were registered. The recordings were made while the participants walked in the central part of the walkway and always in the same direction. We did not record the walking when participants made direction changes (180-degree rotation), because they could result in a possible confounding factor. Indeed, it is well known in the literature that Parkinson's disease affects the ability to maintain dynamic stability during the change of direction [54]. For each subject (in each condition) we collected the best four trials [55–57], in which all the markers were highly visible, and for each trial two gait cycles were selected. Therefore, the evaluation of biomechanical indices was performed on eight gait cycles for each condition. The results were averaged to stabilise the outcome and obtain a more reliable result.

## Acquisition system

The analysis of gait was performed in the Motion Analysis Laboratory of the University of Naples Parthenope. In order to acquire kinematic information of the subjects, we used a stereophotogrammetric system composed of eight infrared cameras (ProReflex Unit—Qualisys Inc., Gothenburg, Sweden) (Fig 1, centre panel) and 55 passive markers (Fig 1, right panel). The markers were positioned in specific anatomical landmarks of each participant in accordance with the modified Davis protocol [58]. Through 3D-GA, we acquired kinematic data useful to calculate the following measures.

## Spatiotemporal parameters

In order to integrate the comparison study on the kinematic indices, we also analysed spatiotemporal gait parameters. In particular, we took into consideration the following parameters: speed (meters/seconds), step length (meters), stance time (seconds), swing time (seconds), cycle time (seconds), double support time (seconds).

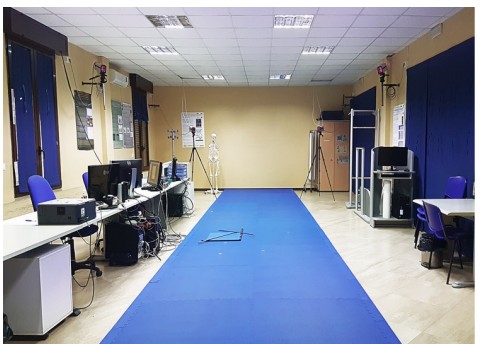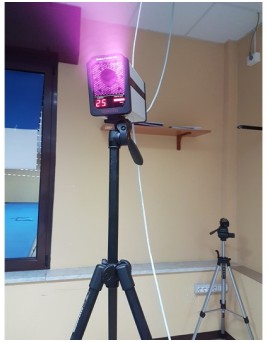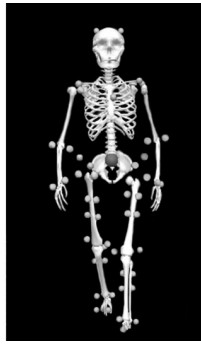

**Fig 1. Acquisition setup.** The left panel shows the acquisition setup. The centre panel shows details of one of the eight cameras employed for the acquisition. The right panel shows a representative image of the patient's acquired data.

**Harmonic ratio.** The HR consists of a spectral analysis of body acceleration, obtained through Fourier transform. This method allows to evaluate the smoothness of gait and estimate the stability, calculating the symmetry within strides [30, 32, 59]. The acceleration components are divided in "in phase" and "out of phase" and correspond respectively to even and odd harmonics. The nature of the acceleration signal in antero-posterior (AP) and vertical (V) direction is biphasic, as each stride consists of two consecutive steps, and the even harmonics amplitude results greater than the odd harmonics one. Hence, the HR in AP and V axes is calculated as the ratio of sum of the amplitude of the even harmonics to the ratio of the sum of the amplitudes of the odd harmonics [60]:

$$HR_{AP,V} = \frac{\Sigma \; Amplitudes \; of \; even \; harmonics}{\Sigma \; Amplitudes \; of \; odd \; harmonics} \tag{1}$$

However, in the medio-lateral (ML) direction the acceleration signal is monophasic as the movement is limb-dependent, and the odd harmonics show greater amplitude compared to even harmonics. Consequently, the HR in ML axis results to be inverted [60]:

$$HR_{ML} = \frac{\Sigma \; Amplitudes \; of \; odd \; harmonics}{\Sigma \; Amplitudes \; of \; even \; harmonics} \tag{2}$$

HR is usually calculated at lower trunk level, as it is the closest external point with respect to the COM [18]. However, through 3D-GA we were able to calculate the position of the COM during walking and measure its real HR.

**Jerk ratio.** The Jerk is the first time derivative of acceleration. It is used to calculate gait parameters related to smoothness of movement [39]. One of the common approaches is to calculate the root mean square (RMS) of the jerk of the body into the three-dimensional space (AP, V and ML axis), in order to obtain a single number (for each axis) representative of the smoothness of movement [61, 62]. Finally, the JR is obtained calculating the logarithmic ratio of ML to V RMS jerk (Eq 3) and AP to V RMS jerk (Eq 4), in order to obtain a dimensionless parameter, expressed in decibel (dB) [39].

$$JR_{AP/V} = 10 log_{10} \left( \frac{RMS \; AP \; Jerk}{RMS \; V \; Jerk} \right) \tag{3}$$

$$JR_{ML/V} = 10 log_{10} \left( \frac{RMS \; ML \; Jerk}{RMS \; V \; Jerk} \right) \tag{4}$$

Authors which used the JR suggested calculating it at head level in order to efficiently show a postural control impairment [26, 39]. Hence, we calculated the JR at head level, using data of the two markers positioned to track the head movements.

**Golden ratio.**   This technique is based on the theory of the golden ratio (*sectio aurea*). In nature there are many phenomena which presents well-known proportions, including ϕ (represented by the number 1.6180...) [41, 42]. This proportion is present when two elements (e.g., *a* and *b*) meet the following criteria:

$$\frac{a+b}{a} = \frac{a}{b} = \phi \qquad (5)$$

Iosa et al. identified the ϕ value in three main proportion of gait phases, defining this measure as harmony of gait. Specifically, the authors defined the golden ratio parameters of gait as the ratio between cycle time and stance time, stance time and swing time, swing time and double support time [29]. In order to obtain a measure representative of the distance between each subject ratio during gait and the ϕ value, we calculated each GR parameter as the absolute value of the difference between the subject ratio and ϕ.

$$GR_1 = \left[ \frac{Cycle\ time}{Stance\ Time} - \phi \right] \qquad (6)$$

$$GR_2 = \left[ \frac{Stance\ time}{Swing\ Time} - \phi \right] \qquad (7)$$

$$GR_3 = \left[ \frac{Swing\ time}{DLS\ Time} - \phi \right] \qquad (8)$$

The pure ratio values have been calculated too in order to perform a correlation analysis similar to Iosa et al. [46].

**Trunk displacement index.**   The TDI was designed keeping into consideration the control exerted by the hierarchically ordered brain structures that, integrating sensory information, are able to control the COM and consequently the balance [63]. To build the index we calculated the ratio between the summation of the norm of the three-dimensional distances between trunk and COM trajectories and the COM mean position:

$$TDI = \frac{\sum \|Td\|}{\sum \|COMd\|} \qquad (9)$$

where Td represents the three-dimensional vector of the distances between trunk trajectory and COM mean position, while COMd represents the three-dimensional vector of the distances between COM trajectory and COM mean position, during gait [21].

## Statistical analysis

The statistical analysis was performed in MATLAB, (Mathworks®, version R2020a). Since the parameters showed a non-normal distribution (after the Shapiro-Wilk test), we performed a two-side Wilcoxon signed rank test in order to compare our data. Test statistic (W) and effect size (ESr) was reported for each comparison [64]. To analyse the relationship between the improvement (difference between OFF and ON condition) of the kinematic and clinical parameters, a correlation test was carried out through a Spearman's correlation analysis. A partial correlation analysis was performed too, in order to control for possible confounding factors. A significance level of $p < 0.05$ has been considered.

**Table 2. Comparison of spatiotemporal parameters before (OFF) and after (ON) L-DOPA intake.**

| Parameter | Mean (±Standard deviation) | | Test Statistic | *p*-value | Effect Size |
|---|---|---|---|---|---|
| | Off | On | | | |
| Speed (m/s) | 0.852 (±0.21) | 1.036 (±0.2) | -231 | < **0.001** | -1 |
| Step Length (m) | 0.506 (±0.12) | 0.569 (±0.09) | -217 | < **0.001** | -0.94 |
| Stance Time (s) | 0.726 (±0.08) | 0.677 (±0.05) | 183 | **0.002** | 0.79 |
| Swing Time (s) | 0.442 (±0.05) | 0.426 (±0.03) | 91 | 0.114 | 0.4 |
| Cycle Time (s) | 1.175 (±0.1) | 1.104 (±0.07) | 191 | < **0.001** | 0.83 |
| Double Support Time (s) | 0.287 (±0.08) | 0.253 (±0.05) | 165 | **0.004** | 0.71 |

The comparison was performed between OFF and ON condition of patients affected by Parkinson's Disease, using a two-tailed Wilcoxon signed rank test. Mean, standard deviation, test statistic, p-value, and effect size values are reported within the table. Units of measurement are meters (m) and seconds (s). Significant p-values in bold.

## Results

### OFF and ON comparison

The UPDRS-III showed a statistical difference before and after L-DOPA intake, where the PDoff patients presented higher UPDRS-III values. With regard to the spatiotemporal parameters, several statistical differences were observed between the OFF and ON condition (Table 2). Specifically, the PDon group showed increased speed (W = -231, p < 0.001, ESr = -1) and step length (W = -217, p < 0.001, ESr = -0.94), and reduced stance time (W = 183, p = 0.002, ESr = 0.79), cycle time (W = 191, p < 0.001, ESr = 0.83), and double support time (W = 165, p = 0.004, ESr = 0.71). Swing time did not show statistically significant variation (W = 91, p = 0.004, ESr = 0.39).

The analysis of the synchrony and smoothness of movement performed through HR in PD patients before and after L-DOPA intake showed a significant difference in one of the three directions. Specifically, the PDoff patients showed lower HR values in the AP axis, compared to PDon patients (W = 119, p = 0.038, ESr = 0.52) (Fig 2). Differently, the smoothness of movement measured through logarithmic dimensionless jerk, failed to show any significant difference between OFF and ON condition, in both $JR_{AP/V}$ (W = -69, p = 0.23, ESr = -0.3) and $JR_{ML/V}$ (W = 29, p = 0.614, ESr = 0.13) measures (Fig 3). The analysis of the fractal harmony, performed through GR, presented significant differences before and after L-DOPA medication, in all three parameters. Specifically, compared to PDon patients, the PDoff patients showed a greater distance from optimal φ values in the $GR_1$ (W = -133, p = 0.021, ESr = -0.58), $GR_2$ (W = -127, p = 0.027, ESr = -0.55) and $GR_3$ (W = -131, p = 0.023, ESr = -0.57) (Fig 4). Finally, analysing the trunk displacement, we found a significant statistical difference between PDon and PDoff (W = -217, p < 0.001, ESr = -0.93), with the PDoff patients exhibiting higher TDI values compared to PDon patients (Fig 5).

### Correlation analysis

Finally, we performed a correlation analysis between the clinical improvement (i.e., the difference between the OFF and ON condition) and the kinematic improvement of the synthetic parameters that showed a significant difference in the OFF-ON comparison. We found a statistically significant correlation between the TDI improvement and the UPDRS-III improvement (r = 0.46; p = 0.035) (Fig 6). Moreover, this result was confirmed even when controlling for the speed improvement, considered as a confounding factor (r = 0.45, p = 0.049). None of the remaining kinematic improvement resulted to be correlated with the clinical

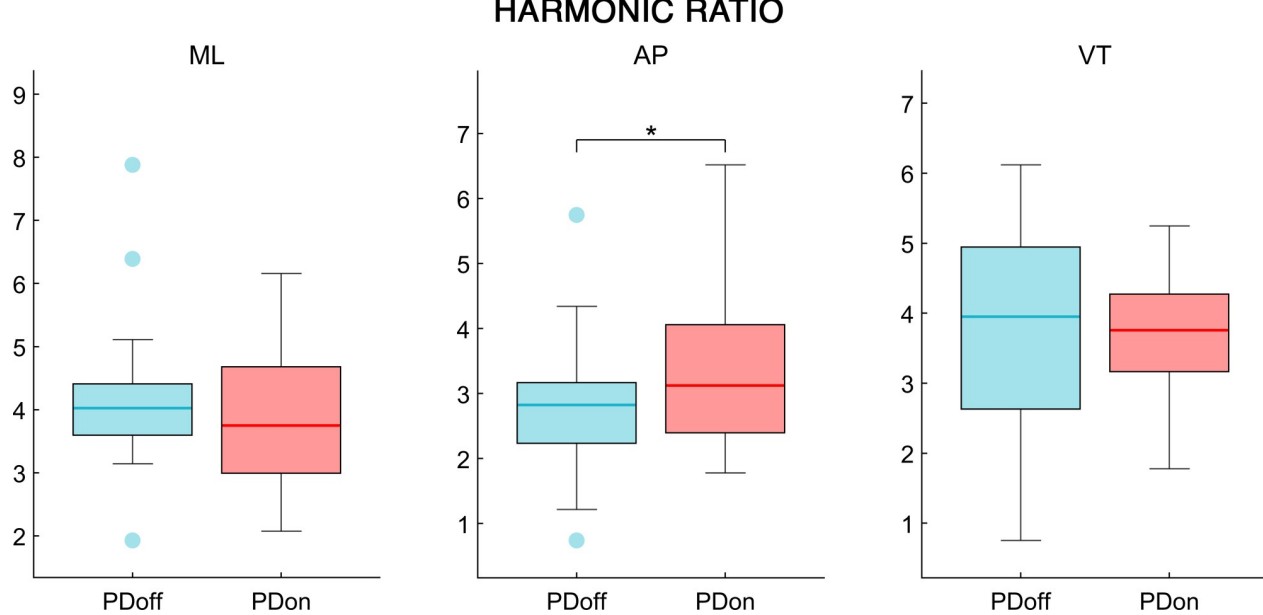

**Fig 2. Harmonic ratio.** The box plot of the harmonic ratio comparison, in mediolateral (ML), anteroposterior (AP) and vertical (VT) directions, between OFF and ON condition in patients affected by Parkinson's disease. The box represents data from 25th to 75th percentiles; the horizontal line inside the box represents the median; lower and upper error lines represent the 10th and 90th percentile respectively; filled circles represent the outliers. Patients affected by Parkinson's disease before L-DOPA intake (PDoff), patients affected by Parkinson's disease after L-DOPA intake (PDon). Significance $p$ value: $^*p < 0.05$, $^{**}p < 0.01$, $^{***}p < 0.001$.

improvement. Furthermore, we performed a correlation analysis between the improvement of the golden ratio parameters as originally calculated by Iosa et al. [46], and the UPDRS-III improvement. Even in this case the correlation did not show any significant result.

## Discussion

In this study we evaluated different synthetic measures of gait, (i.e., HR, JR, GR and TDI) in people affected by PD. Specifically, we measured the responsiveness to L-DOPA intake of those indices, and investigated the relationship between the kinematic and the clinical improvements between the OFF and ON condition.

Firstly, through the spatiotemporal parameters we can observe that after levodopa intake, PD patients increased their walking speed. This resulted in reduced gait cycle duration, and in particular in lower time spent in double support. These results highlight the motor improvement obtained by PD patients in ON condition, and are consistent with previous studies investigating the effects of L-DOPA on PD gait [65–67]. Analysed through HR, PD patients after L-DOPA administration showed significant higher values of HR in the AP direction. This result implies a worse harmony of movement in the PDoff group, which was improved by L-DOPA. The only other study on HR in OFF and ON conditions in PD was performed by Pelicioni et al. [68]. The authors calculated HR at head and pelvis level, showing the effects on different PD subgroups (with and without postural instability and gait difficulty). However, after L-DOPA intake, beyond the PD subtype, results showed increased HR in the AP direction and reduced HR in the VT direction. Even if related to different parts of the body (our HR values are measured at COM level), our result is in agreement with this study, although we failed to prove any effect concerning the HR difference in ML and VT directions. Additional studies comparing HR of the trunk in PD patients and healthy controls showed several

# JERK RATIO

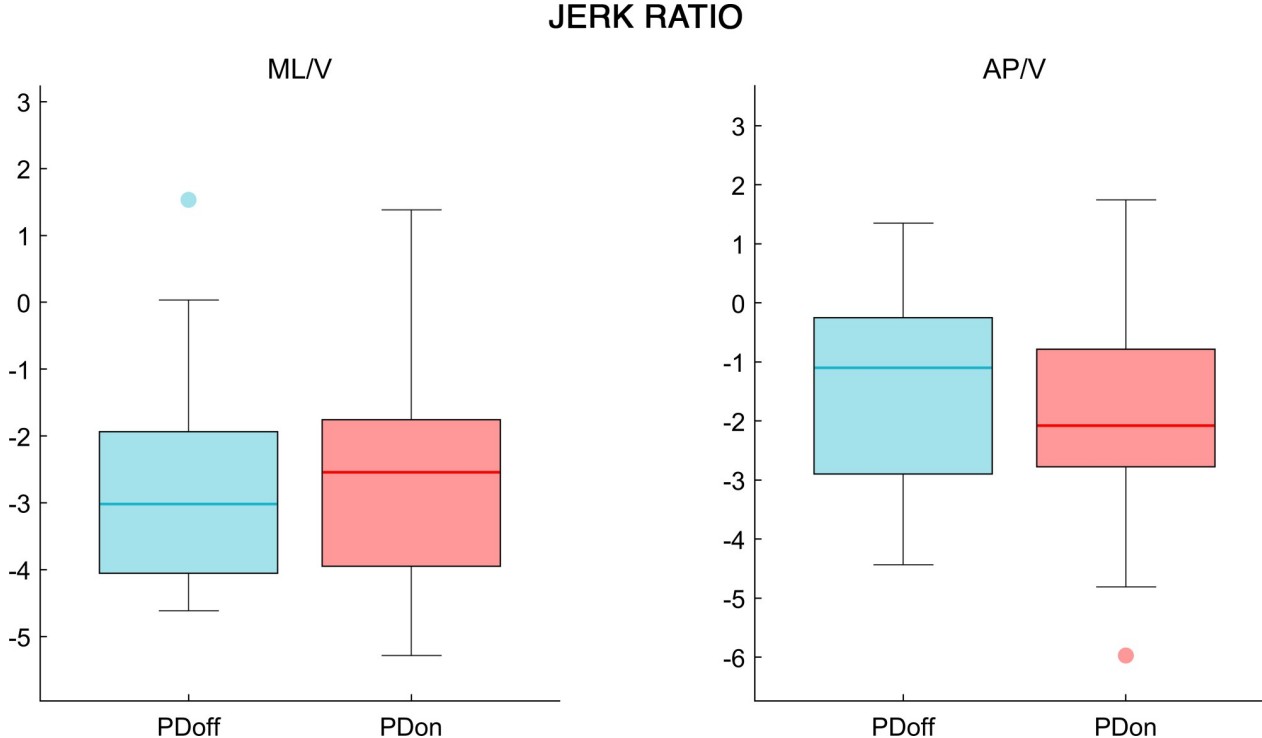

**Fig 3. Jerk ratio.** The box plot of the jerk ratio comparison, in mediolateral to vertical (ML/V) and anteroposterior to vertical (AP/V) ratios, between OFF and ON condition in patients affected by Parkinson's disease. Patients affected by Parkinson's disease before L-DOPA intake (PDoff), patients affected by Parkinson's disease after L-DOPA intake (PDon). Significance $p$ value: $^{*}p < 0.05$, $^{**}p < 0.01$, $^{***}p < 0.001$.

# GOLDEN RATIO

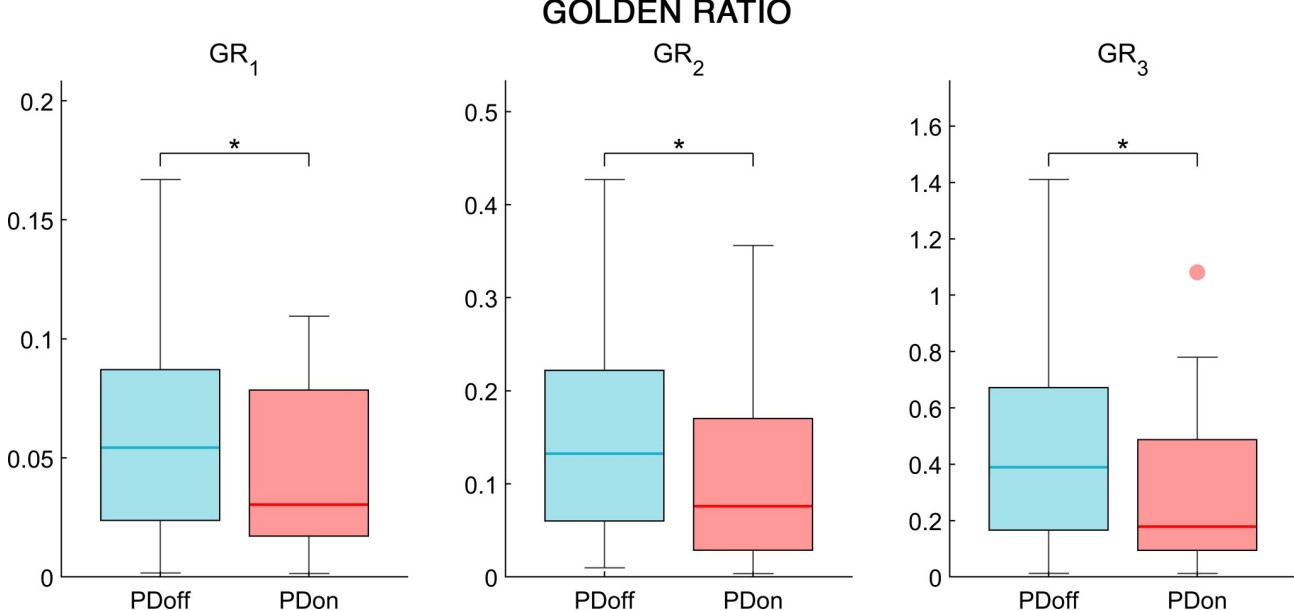

**Fig 4. Golden ratio.** The box plot of the golden ratio comparison, cycle time/stance time ($GR_1$), stance time/swing time ($GR_2$), swing time/double limb support time ($GR_3$), between OFF and ON condition in patients affected by Parkinson's disease. Patients affected by Parkinson's disease before L-DOPA intake (PDoff), patients affected by Parkinson's disease after L-DOPA intake (PDon). Significance $p$ value: $^{*}p < 0.05$, $^{**}p < 0.01$, $^{***}p < 0.001$.

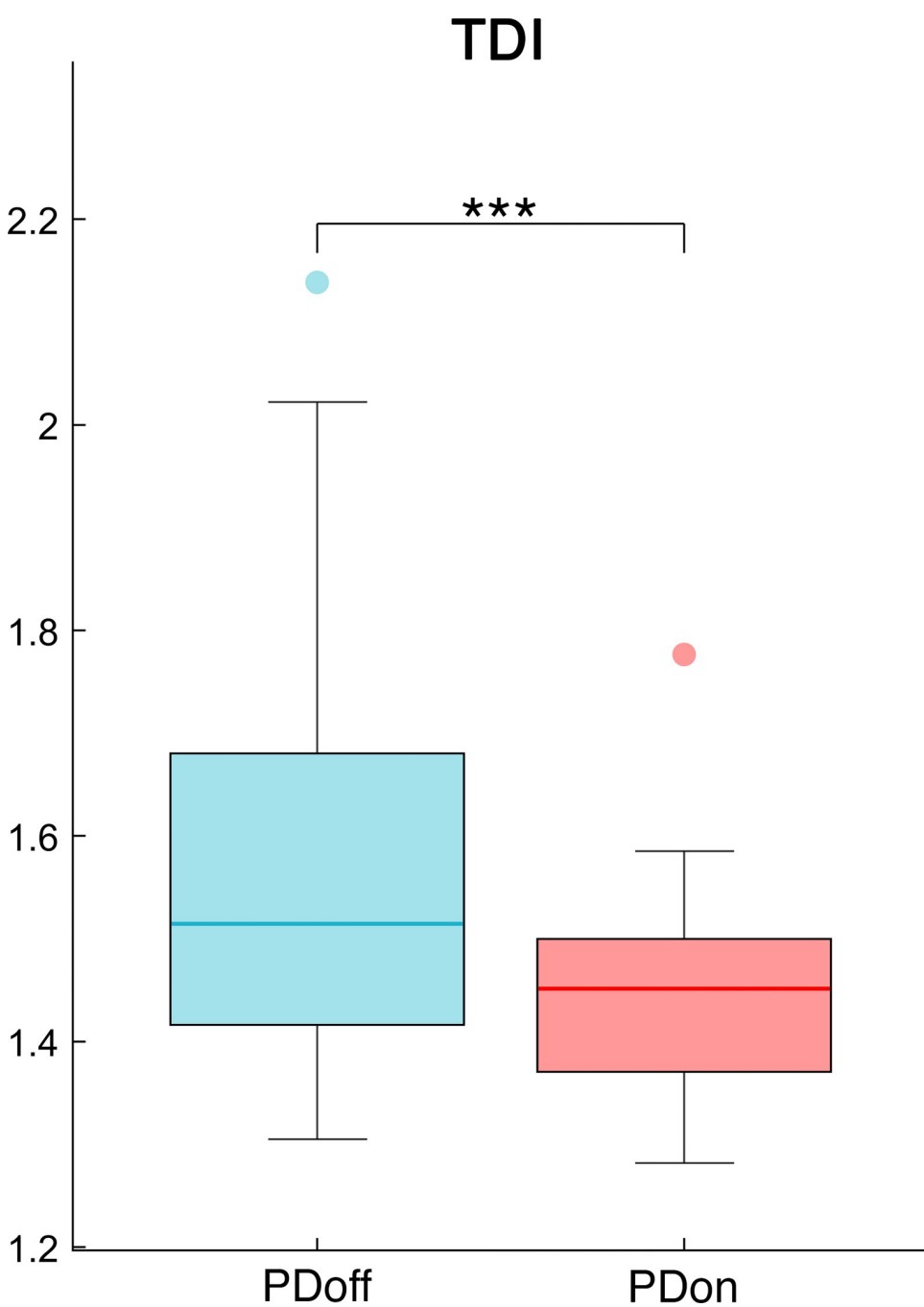

**Fig 5. Trunk displacement index.** The box plot of the Trunk Displacement Index (TDI), between OFF and ON condition in patients affected by Parkinson's disease. Patients affected by Parkinson's disease before L-DOPA intake (PDoff), patients affected by Parkinson's disease after L-DOPA intake (PDon). Significance *p* value: $^*p < 0.05$, $^{**}p < 0.01$, $^{***}p < 0.001$.

discrepancies. Lowry et al. found lower HR in AP and ML directions of individuals with PD, while Buckley et al. only found lower HR in the AP direction of PD patients [25, 26]. Finally, Castiglia et al., and Latt et al. found lower HR in all three axes comparing PD with healthy controls [36, 37]. These differences could be due to the different severity of the disease, but also to

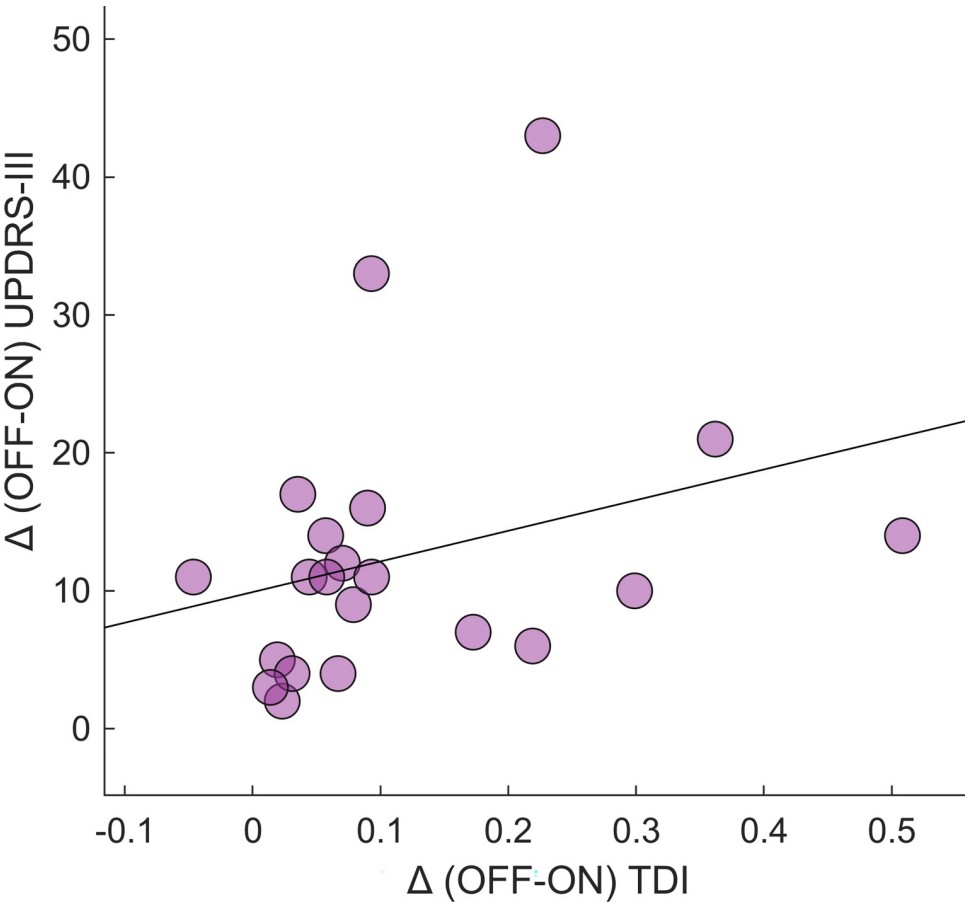

**Fig 6. Correlation between TDI improvement and clinical.** Spearman coefficient correlation between Trunk Displacement Index (TDI) improvement (difference between OFF and ON–Δ (OFF-ON) TDI) and Unified Parkinson's Disease Rating Scale Part III score (UPDRS-III) improvement (Δ (OFF-ON) UPDRS-III). The correlation analysis was performed excluding the effect of the gait speed, considered as a confounding variable. Significance $p$ value: $^{*}p < 0.05$, $^{**}p < 0.01$, $^{***}p < 0.001$.

the methodological approach employed to calculate the HR. Indeed, several studies using inertial sensors estimated that the optimal number of strides to obtain stable HR values is of 20 strides [69–71]. Among the reported studies, only Castiglia et al., and Buckley et al., declared to include at least 20 strides in their analysis. In our case, using a stereophotogrammetric system recording the middle segment of a 10-meters long path, several walking trials were required. Furthermore, each patient underwent two separate recording (OFF and ON phases). Hence, to avoid the effect of the fatigue on the walking performance, we had to reduce the number of recorded trials.

The JR analysis, often considered a measure of the smoothness of gait, could not produce any significant result in our population. The use of a dimensionless logarithmic jerk analysis at head level in PD patients during gait is poorly present in literature. A study performed by Buckley et al. showed that PD individuals presented high values of JR at head level compared to healthy controls [26]. However, no study used this measure to investigate the difference between OFF and ON condition in PD. We consider two possible reasons to explain our result. The first possibility is that the low half dose of L-DOPA that we used was not enough to affect the degree of smoothness given by the variation of acceleration. The second possibility is that

the mechanisms which control the smoothness of movement, related to the rate of change of the acceleration, are non-dopamine dependent, thus we could not observe any difference between OFF and ON conditions.

The GR analysis showed significant differences in $GR_1$, $GR_2$ and $GR_3$. After L-DOPA intake, the PD patients moved their GR values close to the ideal number represented by $\phi$, in each one of the three calculated ratios. L-DOPA was able to improve the ratio between several phases of the gait cycle. Precisely, it improved the ratio cycle time/stance time, stance time/swing time, and swing time/double limb support time. Iosa et al., performing a study on PD and healthy controls showed that individuals with PD during on phase presented GR values farther from $\phi$, when compared to healthy controls [46]. Moreover, after a 12 hours washout the GR values worsened. These results suggest the hypothesis that the harmony of gait, meant as the regulation of the proportion of gait phases, could be influenced by the basal ganglia and thus affected by L-DOPA treatment.

Finally, the TDI showed higher values in PD individuals before L-DOPA intake compared to the same individuals after medication. The TDI was able to differentiate the two conditions, highlighting the postural impairment typical of PD, represented by increased trunk oscillations. As well as for the GR, TDI is a novel measure and it needs to be tested in further and wider population in order to strengthen its validity, although the preliminary results make it a promising biomechanical index.

Concerning the correlation's study, the only measure which showed to be correlated with the UPDRS-III improvements was the TDI. In fact, a positive correlation between the TDI improvement and the clinical motor improvement, evaluated through the UPDRS-III was found. This result contributes to the reliability of the TDI as a useful measure in the assessment of the motor changes induced by the L-DOPA therapy. For both HR and JR, we were not able to find any study which showed a significant correlation with the UPDRS. Concerning GR, Iosa et al., were able to find a significant correlation between each one of the three GR values and the UPDRS [46]. We wondered if the inconsistency between our results and Iosa's could be driven by to the different way we calculated the GR. In their study, the authors used the exact value of the ratio between the gait phases of the subjects [46], while we used the absolute value of the difference between each subject ratio and $\phi$, (as stated in the methods section). We used a different method in order to observe the actual gap between the ideal GR value (i.e., $\phi$) and the one of each subject. Moreover, using the actual ratio values like Iosa et al. we could observe group-averaged values close to $\phi$, while subject-specific values are far from it. Finally, correlating the actual ratios with the UPDRS-III values using linear correlation tests could be misleading, as it would be more consistent for the data to be related through a quadratic correlation. However, in order to verify if the discrepancy of the results could be driven by the way we calculated the parameter, we correlated the UPDRS-III values with the ratio values of the subjects (as performed by Iosa et al.). Nevertheless, even in this case, we could not observe any significant correlation. This difference may be due to the size of our sample or to further characteristics related to our protocol or to our patients, such as the use of a mild L-DOPA dose or the disease severity of the patients. Further studies with increased population might be useful to confirm if a relationship between GR and UPDRS-III exists.

In summary, we observed the response to L-DOPA of synthetic gait indices in an early PD population. The TDI highlighted the presence of trunk impairment in PD, as a consequence of the postural instability typical of the disease, and showed the highest ESr among the measures under consideration in the comparison between OFF phase and ON phase. Moreover, TDI was the only index which showed a significant correlation with the overall motor condition as evaluated by UPDRS-III. From another point of view, the effect of L-DOPA on gait could be observed clearly by an improved proportion between gait cycle phases, measured through GR.

Unfortunately, GR does not offer information concerning the stability of the individual itself or information about the movement in the three directions. However, it offers an evaluation of the gait harmony during the gait cycle and reveals which gait phases should be regulated according to the GR. It is noteworthy that all three investigated GR values improved after the L-DOPA administration, with similar ESr values. Conversely, HR offers information regarding the stability and the smoothness of the walking. L-DOPA effect could be observed on the AP axis of PD patients, which gained better harmony of COM movement. As stated before, JR did not provide any significant result. A limitation of this study is the relatively small sample size. Further studies including a larger population should be carried out to confirm our results. Another limitation of the study is that biomechanical indices analysis was performed on eight gait cycles for each condition, while a higher gait cycle number was used in several studies [69–71]. However, it is important to consider that a higher number of recordings could have fatigued the participants [53], who repeated the analysis twice on the same day before and 40 minutes after the administration of L-DOPA. Finally, it could be useful to perform the same study protocol using different acquisition devices, like inertial sensors [72], and markerless camera systems [73], to evaluate the accuracy required by the tools in order to employ such synthetic indices.

## Conclusions

TDI, GR and HR resulted to be sensitive enough to detect significant difference before and after L-DOPA intake in early PD patients, with the TDI as the only measure which showed a correlation with a clinical parameter. Each measure can be used to analyse a different gait characteristic of individuals affected by PD. TDI, should be employed to evaluate balance and stability through trunk oscillation. GR should be used to evaluate the harmony and the respect of the natural gait phases and proportions. HR should be used to evaluate the smoothness of the COM movement in the three axes of motion.

## Supporting information

**S1 Dataset.**
(XLSX)

## Author Contributions

**Conceptualization:** Emahnuel Troisi Lopez, Roberta Minino, Pierpaolo Sorrentino.

**Funding acquisition:** Giuseppe Sorrentino.

**Investigation:** Emahnuel Troisi Lopez, Roberta Minino, Valentino Manzo, Marianna Liparoti.

**Project administration:** Marianna Liparoti.

**Resources:** Valentino Manzo.

**Software:** Pierpaolo Sorrentino.

**Supervision:** Marianna Liparoti.

**Writing – original draft:** Emahnuel Troisi Lopez, Roberta Minino, Pierpaolo Sorrentino, Marianna Liparoti.

**Writing – review & editing:** Pierpaolo Sorrentino, Domenico Tafuri, Giuseppe Sorrentino, Marianna Liparoti.

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
