## [Decision Letter · Decision Letter 0]

19 Jul 2021

PONE-D-21-17999

Synthetic kinematic indices denoting the overall motor features of gait in Parkinson’s disease: a comparison study

PLOS ONE

Dear Dr. Sorrentino,

Thank you for submitting your manuscript to PLOS ONE. After careful consideration, we feel that it has merit but does not fully meet PLOS ONE’s publication criteria as it currently stands. Therefore, we invite you to submit a revised version of the manuscript that addresses the points raised during the review process.

Two experts in the field have carefully reviewed the manuscript entitled, "Synthetic kinematic indices denoting the overall motor features of gait in Parkinson’s disease: a comparison study". Their comments are appended below. 

Both of them gave favourable comments for publication with leaving several minor concerns including proof reading before submission. 

I will make the final decision after receipt of your necessary revision and the replies to each critiques. 

We look forward to receiving your revised manuscript.

Kind regards,

Manabu Sakakibara, Ph.D.

Academic Editor

PLOS ONE

Journal Requirements:

Reviewers' comments:

Reviewer's Responses to Questions

**Comments to the Author**

1. Is the manuscript technically sound, and do the data support the conclusions?

Reviewer #1: Yes

Reviewer #2: Yes

2. Has the statistical analysis been performed appropriately and rigorously? 

Reviewer #1: Yes

Reviewer #2: Yes

3. Have the authors made all data underlying the findings in their manuscript fully available?

Reviewer #1: Yes

Reviewer #2: Yes

4. Is the manuscript presented in an intelligible fashion and written in standard English?

Reviewer #1: No

Reviewer #2: Yes

5. Review Comments to the Author

Reviewer #1: In this study authors aimed at investigating the synthetic indices which are representative of gait stability and harmony in patients with Parkinson’s disease. In particular they focused on their sensitivity in discriminating the OFF and ON conditions. The paper is interesting and the obtained results could be useful to quantify the severity of motor impairment during gait in patients with Parkinson’s disease, but the following minor issues need to be addressed.

1. The English in the present manuscript requires an improvement. Please carefully proof-read spell check to eliminate grammatical errors. Some periods need also to be revised, like for example the following ones, in the “Introduction” section:

“Gait analysis (GA) is a widely used methodologies to objectively investigate the gait features in health [5] and both non-motor [6,7] and motor diseases, including PD [3,8–11].”

“However, these studies were carried out regardless of the ON or OFF condition and any the comparison between the two states has been performed.”

2. The title should focus on the indices’ sensitivity in discriminating the ON and OFF condition, since it seems to be the main aim of the authors.

3. It would be useful integrating a brief explanation of the meaning of the ON and OFF condition in the Abstract and in the Introduction for casual readers.

4. In the Introduction (Lines 48-49), authors state that through the optoelectronic systems it is possible to acquire spatio-temporal and kinematic parameters with high precision and reliability. Please add some references supporting this statement.

5. Line 49: the sentence “However, more recently, to obtain an overall assessment of gait, many studies have turned to analyse more synthetic…” should be moved in a new paragraph since is not directly linked with the previous topic. Which are the systems commonly used to measure the synthetic indices? Please integrate a description in the Introduction.

6. Table I and patients’ characteristics should be moved in the Subjects section.

7. Intervention section: Why did authors acquire 4 gait cycles? Could they add some reference or explanation?

8. Some pictures of the setup and acquisition should be integrated within the manuscript

9. The quality and resolution of the Figures need to be improved.

10. The limitations and conclusions of the study should be integrated in the paper.

Reviewer #2: This observational study points out the sensitivity of different synthetic indices with regard to clinical motor changes of people with PD. More specifically, the authors have assessed objectively the gait performance of 21 subjects with PD in both OFF and ON phases considering synthetic indices including the Trunk displacement Index (TDI) which is a parameter implemented by the authors in order to assess the relationship between trunk and Center of Mass oscillations. Moreover, the authors investigated whether there is a relationship between these indices and the level of motor impairment assessed by UPDRS - part III under both conditions. The findings showed the sensitivity of TDI as well as HR (Harmony Ratio) only in AP (antero-posterior) axis and GR (Golden ratio) in all three dimensions to ON and OFF conditions in people with PD, while only the TDI was directly correlated with the clinical status of PD subjects. The workflow is of good quality and the English is well written with some small grammatical and punctuation errors. However, some minor concerns should be considered as detailed below.

Main comments

The study design is unclear. Was it a retrospective or a prospective study? The period of patients’ recruitment and the date of study approval by the Ethics Committee are not mentioned in the manuscript. Please clarify these points in order to make your study replicable.

Minor comments

Abstarct

Line 29. Your conclusion relates to a study completed . The past should be used.

Line 42. Please replace “patients’ problems” with other proper terms (i.e: clinical conditions, status, disorders…).

Line 44… GA is a…. methodology…

Line 49. The right term commonly used in GA is: “spatio-temporal” and kinematic parameters.

Line 51. Correct the punctuation error.

Line 52. …individual”s” affected by motor “pathologies”. Please replace with a proper term.

Line 73. Correct the punctuation error.

Line 86. PD Individual”s” or people with PD (the plural should be used).

Line 88. When describing the TDI, information about the score interpretation and its clinical importance lacks. A higher value stands for what?

Line 90. Grammar should be revised.

Line 92. I suggest to rewrite the aim of the study excluding the parts which can be explained in details in the methods (e.g. the number of recruited subjects should be cited in method and results and not among the aim of the study).

Line 99. Was it a retrospective or a prospective study? Please, mention the period of patients’ recruitment.

Line 108. vi) any other …

Line 110. Please report also the date of study approval by the Ethics Committee.

Line 114. I suppose you mean straight “path”.

Intervention

I am wondering how a person with PD can walk 10 meters by 4 gait cycles. This part needs to be clarified. Moreover, can the authors explain if only one trial consisting of four gait cycles were taken into account per condition or a mean of more trial repetitions was considered for the data analysis?

Line 223. Please add that this result is significant. The statistical differences can be also non-significant.

Line 323. “Summerising” is not in line with the English style of the whole text.

6. PLOS authors have the option to publish the peer review history of their article (what does this mean?). If published, this will include your full peer review and any attached files.

Reviewer #1: **Yes: **Erika D'Antonio

Reviewer #2: **Yes: **Sanaz Pournajaf

---

## [Author Response · Author response to Decision Letter 0]

13 Aug 2021

Dear Manabu Sakakibara,

please find enclosed a revision of our paper entitled “Sensitivity to gait improvement after levodopa intake in Parkinson’s disease: a comparison study among synthetic kinematic indices.” to be considered for publication in Plos One.

We thank the editor and the reviewers for their efforts in evaluating our manuscript. We did our best to follow the comments made by the reviewers, and hope that this revised submission will be adequate for publication.

In what follows, we firstly summarize the main changes, and then provide detailed answers to Reviewers’ comments.

Main changes:

General: the English have been revised throughout the whole manuscript. A new figure was added; the figures have been re-numbered and re-uploaded.

Title: the title has been modified according to Reviewer #1 in order to better focus on the aim of the study. Consequently, the new title is: “Sensitivity to gait improvement after levodopa intake in Parkinson’s disease: a comparison study among synthetic kinematic indices”.

Abstract: we integrated a brief explanation concerning the OFF and ON (medication) state of the individuals affected by Parkinson’s disease.

Introduction: according to the Reviewers, missing information have been updated in order to enhance the study background. The aims have been improved removing unnecessary information.

Methods: following Reviewers’ observation, methods have been updated to better clarify our study design and intervention. Subject’s data table have been included in the correct section.

Results: the table containing subjects’ data has been removed from this section.

Discussion: limitations have been added and the conclusions have been moved to a separate section.

Review Comments to the Author

Reviewer #1: In this study authors aimed at investigating the synthetic indices which are representative of gait stability and harmony in patients with Parkinson’s disease. In particular they focused on their sensitivity in discriminating the OFF and ON conditions. The paper is interesting and the obtained results could be useful to quantify the severity of motor impairment during gait in patients with Parkinson’s disease, but the following minor issues need to be addressed.

1.Reviewer: The English in the present manuscript requires an improvement. Please carefully proof-read spell check to eliminate grammatical errors. Some periods need also to be revised, like for example the following ones, in the “Introduction” section:

“Gait analysis (GA) is a widely used methodologies to objectively investigate the gait features in health [5] and both non-motor [6,7] and motor diseases, including PD [3,8–11].”

“However, these studies were carried out regardless of the ON or OFF condition and any the comparison between the two states has been performed.”

1.Authors: Thank you for this suggestion. A grammar check has been carried out and errors have been corrected throughout the whole manuscript. The two above-mentioned sentences (lines 50-52 and line 99, respectively) have been modified.

2.Reviewer: The title should focus on the indices’ sensitivity in discriminating the ON and OFF condition, since it seems to be the main aim of the authors.

2.Authors: According to the reviewer's suggestion, the title has been changed to:

“Sensitivity to gait improvement after levodopa intake in Parkinson’s disease: a comparison study among synthetic kinematic indices.”

3.Reviewer: It would be useful integrating a brief explanation of the meaning of the ON and OFF condition in the Abstract and in the Introduction for casual readers.

3.Authors: thanks for this recommendation. A brief explanation of the OFF and ON conditions has been included in the abstract (lines 23-24) and in the manuscript (lines 44-46), in order to make the paper clearer for casual readers.

4.Reviewer: In the Introduction (Lines 48-49), authors state that through the optoelectronic systems it is possible to acquire spatio-temporal and kinematic parameters with high precision and reliability. Please add some references supporting this statement.

4.Authors: As requested, we added some references to support our statement (line 56).

New References:

Ceseracciu E, Sawacha Z, Cobelli C. Comparison of markerless and marker-based motion capture technologies through simultaneous data collection during gait: proof of concept. PLoS One. 2014;9(3):e87640. 

Fusca M, Negrini F, Perego P, Magoni L, Molteni F, Andreoni G. Validation of a wearable IMU system for gait analysis: Protocol and application to a new system. Appl Sci. 2018;8(7):1167. 

Anwary AR, Yu H, Callaway A, Vassallo M. Validity and consistency of concurrent extraction of gait features using inertial measurement units and motion capture system. IEEE Sens J. 2020;21(2):1625–34. 

5.Reviewer: Line 49: the sentence “However, more recently, to obtain an overall assessment of gait, many studies have turned to analyse more synthetic…” should be moved in a new paragraph since is not directly linked with the previous topic. Which are the systems commonly used to measure the synthetic indices? Please integrate a description in the Introduction.

5.Authors: The sentence has been moved as suggested (line 57). In addition, systems that are usually employed to calculate synthetic measures have been included as well (lines 59-60).

6.Reviewer: Table I and patients’ characteristics should be moved in the Subjects section.

6.Authors: As suggested by the reviewer, the Table I has been moved under the Subject session (line 124)

7.Reviewer: Intervention section: Why did authors acquire 4 gait cycles? Could they add some reference or explanation?

7.Authors: We thank the Reviewer for the opportunity to better explain the intervention section. Patients were recorded while walking forth and back at self-selected speed, along a 10-meter walkway. In accordance with the literature (references added at line 148), for each subject and in each condition (OFF and ON), we collected the four best trials and for each of these two gait cycles were selected. The two gait cycles were chosen in the central part of recordings where the markers are highly visible, and the gait was not affected by the possible confounding factors related to the trajectory’s changes. Subsequently the average of eight gait cycles data (two gait cycles for four trials) was used to stabilise the outcome and obtain a more reliable result. Intervention section (line 136) has been updated.

New References:

Nelson AJ, Zwick D, Brody S, Doran C, Pulver L, Rooz G, et al. The validity of the GaitRite and the Functional Ambulation Performance scoring system in the analysis of Parkinson gait. NeuroRehabilitation. 2002;17(3):255–62. 

Agosti V, Vitale C, Avella D, Rucco R, Santangelo G, Sorrentino P, et al. Effects of Global Postural Reeducation on gait kinematics in parkinsonian patients: a pilot randomized three-dimensional motion analysis study. Neurol Sci. 2016;37(4):515–22. 

Galafate D, Pournajaf S, Condoluci C, Goffredo M, Di Girolamo G, Manzia CM, et al. Bilateral foot orthoses elicit changes in gait kinematics of adolescents with down syndrome with flatfoot. Int J Environ Res Public Health. 2020;17(14):4994. 

8.Reviewer: Some pictures of the setup and acquisition should be integrated within the manuscript

8.Authors: Thank you for this suggestion. Figure 1 has been added (line 152) representing the setup, the camera details and the acquisition representation.

9.Reviewer: The quality and resolution of the Figures need to be improved.

9.Authors: The figures have been re-uploaded with higher quality and a 600dpi resolution (please download it for full quality; preview is low quality only).

10.Reviewer: The limitations and conclusions of the study should be integrated in the paper.

10.Authors: Thank you for your suggestion. The limitations of the study have been included in the final part of the discussion (lines 372-376), as well as the conclusions paragraph (line 376).

New References:

Cuesta-Vargas AI, Galán-Mercant A, Williams JM. The use of inertial sensors system for human motion analysis. Phys Ther Rev. 2010;15(6):462–73. 

D’Antonio E, Taborri J, Mileti I, Rossi S, Patané F. Validation of a 3D Markerless System for Gait Analysis based on OpenPose and Two RGB Webcams. IEEE Sens J. 2021; 

Reviewer #2: This observational study points out the sensitivity of different synthetic indices with regard to clinical motor changes of people with PD. More specifically, the authors have assessed objectively the gait performance of 21 subjects with PD in both OFF and ON phases considering synthetic indices including the Trunk displacement Index (TDI) which is a parameter implemented by the authors in order to assess the relationship between trunk and Center of Mass oscillations. Moreover, the authors investigated whether there is a relationship between these indices and the level of motor impairment assessed by UPDRS - part III under both conditions. The findings showed the sensitivity of TDI as well as HR (Harmony Ratio) only in AP (antero-posterior) axis and GR (Golden ratio) in all three dimensions to ON and OFF conditions in people with PD, while only the TDI was directly correlated with the clinical status of PD subjects. The workflow is of good quality and the English is well written with some small grammatical and punctuation errors. However, some minor concerns should be considered as detailed below.

Main comments

1.Reviewer: The study design is unclear. Was it a retrospective or a prospective study? The period of patients’ recruitment and the date of study approval by the Ethics Committee are not mentioned in the manuscript. Please clarify these points in order to make your study replicable.

1.Authors: Thank you for this comment. This is a mainly observational study, because we performed the gait analysis just before and 40 minutes after levodopa medication; this was clarified in the manuscript within the intervention section (line 136). Moreover, we included the date of study approval by the Ethics Committee (line 122) and specified the period of patients’ recruitment (lines 112-113).

Minor comments

2.Reviewer: Abstract - Line 29. Your conclusion relates to a study completed . The past should be used.

2.Authors: Thank you. We changed it, as suggested (line 33).

3.Reviewer: Line 42. Please replace “patients’ problems” with other proper terms (i.e: clinical conditions, status, disorders…).

3.Authors: Corrected as suggested (line 48).

4.Reviewer: Line 44… GA is a…. methodology…

4.Authors: Corrected (line 50).

5.Reviewer: Line 49. The right term commonly used in GA is: “spatio-temporal” and kinematic parameters.

5.Authors: Modified (line 56).

6.Reviewer: Line 51. Correct the punctuation error.

6.Authors: We checked and corrected the error (line 59).

7.Reviewer: Line 52. …individual”s” affected by motor “pathologies”. Please replace with a proper term.

7.Authors: Replaced with “movement disorders” (line 61).

8.Reviewer: Line 73. Correct the punctuation error.

8.Authors: Corrected (line 82).

9.Reviewer: Line 86. PD Individual”s” or people with PD (the plural should be used).

9.Authors: Corrected (line 97).

10.Reviewer: Line 88. When describing the TDI, information about the score interpretation and its clinical importance lacks. A higher value stands for what?

10.Authors: We apologise for this missing information. The interpretation has been added (higher TDI values = worse stability) at lines 91-92.

11.Reviewer: Line 90. Grammar should be revised.

11.Authors: We rephrased the sentence in a more correct and clear way (lines 100-101).

12.Reviewer: Line 92. I suggest to rewrite the aim of the study excluding the parts which can be explained in details in the methods (e.g. the number of recruited subjects should be cited in method and results and not among the aim of the study).

12.Authors: Thank you for the suggestion. We removed unnecessary information from this section and (hopefully) improved the readability(lines 101-105).

13.Reviewer: Line 99. Was it a retrospective or a prospective study? Please, mention the period of patients’ recruitment.

13.Authors: Our study is mainly observational. The patients’ recruitment period has been added as well (lines 112-113).

14.Reviewer: Line 108. vi) any other …

14.Authors: Corrected (lines 118-119).

15.Reviewer: Line 110. Please report also the date of study approval by the Ethics Committee.

15.Authors: We added the date at line 122.

16.Reviewer: Line 114. I suppose you mean straight “path”.

16.Authors: Exactly, thank you for noticing it (line 137).

17.Reviewer: Intervention - I am wondering how a person with PD can walk 10 meters by 4 gait cycles. This part needs to be clarified. Moreover, can the authors explain if only one trial consisting of four gait cycles were taken into account per condition or a mean of more trial repetitions was considered for the data analysis?

17.Authors: We apologise for being unclear. Individuals (PD patients at very early stage of the disease) walked freely back and forth along a 10-metre long straight walkway. They walked at their own speed and preferred cadence, and we recorded the gait cycles along the entire walkway. After finishing the gait recording, we considered the best four trials and selected two gait cycles (for each trial) in the central area of the walkway, where all markers were visible to the cameras and where the gait was not influenced by turns and changes of direction. The results of the 8 gait cycles (2 cycles for 4 trials) were averaged to obtain more reliable data. Then the patient took his medication and we waited for it to make effect in order to repeat the same acquisition while in ON condition.

The explanation has been hopefully improved at lines 145-149 and references have been added.

New References:

Nelson AJ, Zwick D, Brody S, Doran C, Pulver L, Rooz G, et al. The validity of the GaitRite and the Functional Ambulation Performance scoring system in the analysis of Parkinson gait. NeuroRehabilitation. 2002;17(3):255–62. 

Agosti V, Vitale C, Avella D, Rucco R, Santangelo G, Sorrentino P, et al. Effects of Global Postural Reeducation on gait kinematics in parkinsonian patients: a pilot randomized three-dimensional motion analysis study. Neurol Sci. 2016;37(4):515–22. 

Galafate D, Pournajaf S, Condoluci C, Goffredo M, Di Girolamo G, Manzia CM, et al. Bilateral foot orthoses elicit changes in gait kinematics of adolescents with down syndrome with flatfoot. Int J Environ Res Public Health. 2020;17(14):4994. 

18.Reviewer: Line 223. Please add that this result is significant. The statistical differences can be also non-significant.

18.Authors: Thank you for this comment. We modified it as suggested (lines 255 and 259).

19.Reviewer: Line 323. “Summerising” is not in line with the English style of the whole text.

19.Authors: We modified it with “In summary” (line 360).

---

## [Decision Letter · Decision Letter 1]

18 Oct 2021

PONE-D-21-17999R1Sensitivity to gait improvement after levodopa intake in Parkinson’s disease: a comparison study among synthetic kinematic indicesPLOS ONE

Dear Dr. Sorrentino,

Thank you for submitting your manuscript to PLOS ONE. After careful consideration, we feel that it has merit but does not fully meet PLOS ONE’s publication criteria as it currently stands. Therefore, we invite you to submit a revised version of the manuscript that addresses the points raised during the review process.

Newly participated reviewers are carefully reviewed the revision. Their comments are appended below.

The reviewer 3 pointed out that the Discussion section should be rewritten according to the previous literature on the same topics. Another serious concerns are raised through the manuscript as mentioned in the review comments to the authors.

The reviewer #4 is satisfied with the revised manuscript.

Thus this Academic Editor judged the revised manuscript required modification according to critiques by reviewers.

I will make the final judgement after receipt of each replies and necessary revision.

We look forward to receiving your revised manuscript.

Kind regards,

Manabu Sakakibara, Ph.D.

Academic Editor

PLOS ONE

Reviewers' comments:

Reviewer's Responses to Questions

**Comments to the Author**

1. If the authors have adequately addressed your comments raised in a previous round of review and you feel that this manuscript is now acceptable for publication, you may indicate that here to bypass the “Comments to the Author” section, enter your conflict of interest statement in the “Confidential to Editor” section, and submit your "Accept" recommendation.

Reviewer #3: (No Response)

Reviewer #4: All comments have been addressed

2. Is the manuscript technically sound, and do the data support the conclusions?

Reviewer #3: Partly

Reviewer #4: Yes

3. Has the statistical analysis been performed appropriately and rigorously? 

Reviewer #3: N/A

Reviewer #4: Yes

4. Have the authors made all data underlying the findings in their manuscript fully available?

Reviewer #3: No

Reviewer #4: Yes

5. Is the manuscript presented in an intelligible fashion and written in standard English?

Reviewer #3: Yes

Reviewer #4: Yes

6. Review Comments to the Author

Reviewer #3: Introduction

The introduction is not updated according to the recent scintific literature on the same topic.

Methods

Line 145-150. Many acceleration-derived gait indices require a number of strides > 10 to reach good reliability in their calculations (Riva et al., 2014; Kroneberg et al., 2019). In particular, to calculate the harmonic ratios, 20 harmonics and 20 consecutive gait strides at least are recommended (Pasciuto et al., 2017). This is clearly not possible in a 10-meters pathway. How did you manage this issue? Moreover, in a recent study (Castiglia et al., 2021), HR values in the antero-posterior direction < 1.50 showed to characterize the gait alteration of subjects with PD in ON condition, as respect to healthy subjects. In figure 2, HR values > 4 are reported for subjects with PD. This is probably due to the low number of consecutive recorded strides.

Line 226-227. Did you perform the correlation analysis between the punctual values before and after the levodopa intake, or did you perform it between the improvements (the differences between the ON phase and the OFF phase)?

Results.

In the results section, no information on the spatio-temporal parameters have been provided. Since spatio-temporal gait parameters, such as gait speed and step length, are known to improve after levodopa intake (Bryant et al., 2011; Smulders et al., 2016; Baudentistel et al., 2021), spatio temporal parameters should be reported to improve the interpretability of the results. As regard the correlation analysis, if the aim of the study was to observe the sensitivity-to-change of the indices after the levodopa intake, a correlation analysis between the improvements in the kinematic and the clinical variables should be reported. Moreover, since UPDRS improvements are correlated with the improvements in gait speed after levodopa intake, it would be relevant to exclude the effects of the gait speed improvements between in the correlation analysis.

Discussion.

L 304-306. I believe that this paragraph needs to be improved. I don’t really believe that the authors assessed the sensitivity or the ability to discriminate between on and off conditions. To do that, the authors should have performed other statistical analysis, such as Mann-Whitney test using the medication condition as between factor, AUC calculations, sensitivity and specificity calculations at the optimal cutoff points the discriminate between the two phases. I believe that, in this study, the Authors have rather observed the responsiveness to levodopa intake of the synthetic kinematic indices, in terms of sensitivity-to-change. For this purpose, the authors should focus on the effect size values, to describe the magnitude of the modifications after the levodopa intake.

L352-354. The authors should explain the reason for using another method than the one used by Iosa et al.

L 354-356. This information is not reported in the results section nor in the methods section. Consider adding it in the previous sections or removing the sentence from this section.

L357-358. Please, explain which differences in the subjects’ characteristics could have led to the differences between your study and the one from Iosa et al.

Again, the discussion is not based on the previous literature on the same topic (indexes of gait stability in Parkinson disease)

Reviewer #4: I enjoyed reading this revised version of the manuscript. I think the authors did a good job addressing the useful reviewers' comment.

7. PLOS authors have the option to publish the peer review history of their article (what does this mean?). If published, this will include your full peer review and any attached files.

Reviewer #3: No

Reviewer #4: No

---

## [Author Response · Author response to Decision Letter 1]

13 Nov 2021

Dear Manabu Sakakibara,

please find enclosed a revision of our paper entitled “Sensitivity to gait improvement after levodopa intake in Parkinson’s disease: a comparison study among synthetic kinematic indices” to be considered for publication in Plos One.

We thank the editor and the reviewers for their efforts in evaluating our manuscript. We did our best to follow the received comments, and hope that this revised submission will be adequate for publication.

In what follows, we firstly summarize the main changes, and then provide detailed answers to Reviewers’ comments.

Main changes:

Abstract: after reviewing the manuscript, the abstract has been updated accordingly.

Introduction: according to the Reviewers, the literature has been updated in order to include recent information on the use of synthetic kinematic indices in Parkinson’s disease.

Methods: methods have been updated to include a clearer explanation of the recording method, and the calculated parameters.

Results: results and table concerning the spatiotemporal parameters have been added. The section on the correlation analysis has been totally rewritten in accordance with the reviewers’ suggestion (Fig 6 and caption changed accordingly).

Discussion: comparison with the existing literature has been improved to include more recent literature. New discussion on the spatiotemporal parameters has been included. The discussion on the correlation analysis has been rewritten. Limitations have been enriched in accordance with the concerning expressed by the reviewers.

In addition, we sent an excel file containing main data supporting the results of this study.

Reviewers’ comments

Reviewer #3:

INTRODUCTION

1.Reviewer: The introduction is not updated according to the recent scientific literature on the same topic.

1.Authors: We are sorry for this missing. An update of the literature has been integrated into the introduction of the manuscript.

METHODS

2a.Reviewer: Line 145-150. Many acceleration-derived gait indices require a number of strides > 10 to reach good reliability in their calculations (Riva et al., 2014; Kroneberg et al., 2019). In particular, to calculate the harmonic ratios, 20 harmonics and 20 consecutive gait strides at least are recommended (Pasciuto et al., 2017). This is clearly not possible in a 10-meters pathway. How did you manage this issue?

2a.Authors: We apologize for being unclear and thank the reviewers for the opportunity to better explain the methods. As specified in the manuscript (lines 160-170), patients were asked to walk back and forth continuously along a 10-meter walkway. This allowed us to make at least six gait recordings, from which it was possible to extrapolate four good trials, each of which contained two gait cycles (REF: Nelson AJ et al. The validity of the GaitRite and the Functional Ambulation Performance scoring system in the analysis of Parkinson gait; Agosti V et al. Effects of Global Postural Reeducation on gait kinematics in parkinsonian patients: a pilot randomized three-dimensional motion analysis study). In total eight gait cycles for subjects and conditions were used for the statistical analysis. We chose to record the walk only when patients walked in the centre of the walkway, avoiding the recording of direction changes, which represent one of the main motor impediments of patients with Parkinson's disease and which could compromise dynamic stability (REF: Walking Along Curved Trajectories. Changes With Age and Parkinson's Disease. Hints to Rehabilitation). We understand the Reviewers' objection regarding the reduced number of gait cycles used for synthetic index analysis, however it must be considered that patients performed the task twice in the same day in two different conditions, before and 40 minutes after taking L-DOPA. The number of records used for our analysis of the synthetic indices represents the only possible trade off to avoid participants fatigue that inevitably would affect the gait kinematics (REF: Fatigue in Parkinson's disease: A systematic review and meta-analysis). However, in accordance with the reviewers' suggestion we wrote that one of the limitations of our study is the number of gait cycles used for the analysis of the synthetic indices.

2b.Reviewer: Moreover, in a recent study (Castiglia et al., 2021), HR values in the antero-posterior direction < 1.50 showed to characterize the gait alteration of subjects with PD in ON condition, as respect to healthy subjects. In figure 2, HR values > 4 are reported for subjects with PD. This is probably due to the low number of consecutive recorded strides.

2b.Authors: We want to highlight that HR values are actually calculated at the level of the centre of mass (COM). Most of the studies which use the HR parameters, use data collected by accelerometer positioned at the level of the lower trunk (including Castiglia et al., 2021), as it is the closest external anatomical point with respect to the COM (Iosa et al., 2016, Simoni et al., 2021, Iosa et al., 2013, Fusca et al., 2018, Siragy and Nantel, 2018). Through a stereophotogrammetric system it is possible to retrieve information on the position of the COM itself. It is true that when a subject is at rest in a standing posture, the L3-L5 vertebrae position is the closest point with respect to the COM, but during movement, COM position can change this distance, since its position depends on the position of each body element. Hence, we expect that parameters calculated at L3-L5 level would not be of the same size as the ones calculated at COM level.

However, we calculated the antero-posterior HR values at pelvis level to verify the agreement with the results reported in Castiglia et al., 2021. The authors reported antero-posterior HR values equal to 2.00 (+/-0.55) for Hoehn and Yahr (HY) staging = 1; HR AP 1.94 (+/-0.51) for HY staging = 2; HR AP 1.65 (+/-0.34) for HY staging = 3. Our sample included patients with HY staging ≤ 3 and displayed a value of antero-posterior HR equal to 2.02 (+/-0.41). Hence, pelvis level, our results are similar to the ones reported by Castiglia et al., therefore, the difference highlighted by the reviewer may be driven by the different anatomical position took into account.

3.Reviewer: Line 226-227. Did you perform the correlation analysis between the punctual values before and after the levodopa intake, or did you perform it between the improvements (the differences between the ON phase and the OFF phase)?

3.Authors: This is an excellent observation. We performed the correlation using the punctual values corresponding to the OFF and the ON conditions. In the new revised manuscript, in accordance with the expressed concerns and suggestions, we performed the correlation between the improvements (i.e., the differences between the ON phase and the OFF phase). The previous correlation analysis has been removed from the manuscript. Fig 6 and caption have been updated accordingly. Methods updated at L260-261.

RESULTS

3.Reviewer: In the results section, no information on the spatio-temporal parameters have been provided. Since spatio-temporal gait parameters, such as gait speed and step length, are known to improve after levodopa intake (Bryant et al., 2011; Smulders et al., 2016; Baudentistel et al., 2021), spatio temporal parameters should be reported to improve the interpretability of the results.

3.Authors: Thanks for this recommendation. In order to facilitate the readability and the interpretability of the manuscript, we reported the information concerning spatiotemporal parameters. Mean, standard deviation, and statistical comparison results (p-value, test statistic and effect size) were reported for: speed, step length, stance time, swing time, cycle time, double support time. Methods updated at L191-195. Results updated at L270-282. Discussion updated L355-359.

4.Reviewer: As regard the correlation analysis, if the aim of the study was to observe the sensitivity-to-change of the indices after the levodopa intake, a correlation analysis between the improvements in the kinematic and the clinical variables should be reported.

4.Authors: As aforementioned, following the suggestions, we performed a correlation analysis between the improvements of the kinematic indices and the clinical variable (UPDRS). Results reported at L323-346.

5.Reviewer: Moreover, since UPDRS improvements are correlated with the improvements in gait speed after levodopa intake, it would be relevant to exclude the effects of the gait speed improvements between in the correlation analysis.

5.Authors: We improved our analysis by repeating the correlation test between synthetic indices improvement and clinical improvement, controlling for the effect of the gait speed. Even in this case, the correlation between the TDI improvement and the UPDRS improvement resulted to be statistically significant. Methods updated at L262-263. Results updated at L323-346.

DISCUSSION

6.Reviewer: L 304-306. I believe that this paragraph needs to be improved. I don’t really believe that the authors assessed the sensitivity or the ability to discriminate between on and off conditions. To do that, the authors should have performed other statistical analysis, such as Mann-Whitney test using the medication condition as between factor, AUC calculations, sensitivity and specificity calculations at the optimal cutoff points the discriminate between the two phases. I believe that, in this study, the Authors have rather observed the responsiveness to levodopa intake of the synthetic kinematic indices, in terms of sensitivity-to-change. For this purpose, the authors should focus on the effect size values, to describe the magnitude of the modifications after the levodopa intake.

6.Authors: Thanks for this highly pertinent observation. We have used the term “sensitivity” absolutely incorrectly. As the reviewer rightly states, in our work we evaluate the responsiveness to levodopa of the kinematic synthetic indices under consideration. Accordingly, we revised the discussion section.

7.Reviewer: L352-354. The authors should explain the reason for using another method than the one used by Iosa et al.

7.Authors: We reported the explanation at lines 417-423. We wanted to observe how much each individual golden ratio (GR) was far from phi. Using the Iosa et al. method, a group of subjects could present a mean GR extremely close to phi, but the specific values of GR of each individual are very far from phi. Conversely, another group of people may have a mean GR less close to phi compared to the previous hypothetical group, but in this case each subject presents an individual GR close to phi. Paradoxically, the first group would result to have the best GR value (with higher standard deviation of course), while the truth is that the second group presents values closer to phi. However, the main issue occurs when performing correlation tests. Performing a linear correlation would not be correct in this case because the individuals are logically expected to show worse condition (high UPDRS) when GR is very low compared to phi, better condition (low UPDRS) when GR is close to phi, and again worse condition (high UPDRS) when GR is very high compared to phi. In this case the relationship between variables should be quadratic and cannot be represented by a linear correlation. The way we calculated the GR values, as showed in the method section allows to use linear correlation tests.

8.Reviewer: L 354-356. This information is not reported in the results section nor in the methods section. Consider adding it in the previous sections or removing the sentence from this section.

8.Authors: We updated the methods (L243-244) and the results (L333-335) section in order to report the described result.

9.Reviewer: L357-358. Please, explain which differences in the subjects’ characteristics could have led to the differences between your study and the one from Iosa et al.

9.Authors: We reported the information as requested in L426-428.

10.Reviewer: Again, the discussion is not based on the previous literature on the same topic (indexes of gait stability in Parkinson disease).

10.Authors: We updated the discussion with more recent literature concerning the use of synthetic gait indices in Parkinson’s disease.

Reviewer #4:

Reviewer: I enjoyed reading this revised version of the manuscript. I think the authors did a good job addressing the useful reviewers' comment.

Authors: We are glad the reviewer enjoyed the reading of the manuscript. We thank the reviewer for his effort.

---

## [Decision Letter · Decision Letter 2]

29 Apr 2022

Sensitivity to gait improvement after levodopa intake in Parkinson’s disease: a comparison study among synthetic kinematic indices

PONE-D-21-17999R2

Dear Dr. Sorrentino,

We’re pleased to inform you that your manuscript has been judged scientifically suitable for publication and will be formally accepted for publication once it meets all outstanding technical requirements.

Kind regards,

J. Lucas McKay, Ph.D., M.S.C.R.

Academic Editor

PLOS ONE

Additional Editor Comments (optional):

Reviewers' comments:

Reviewer's Responses to Questions

**Comments to the Author**

1. If the authors have adequately addressed your comments raised in a previous round of review and you feel that this manuscript is now acceptable for publication, you may indicate that here to bypass the “Comments to the Author” section, enter your conflict of interest statement in the “Confidential to Editor” section, and submit your "Accept" recommendation.

Reviewer #3: All comments have been addressed

2. Is the manuscript technically sound, and do the data support the conclusions?

Reviewer #3: Partly

3. Has the statistical analysis been performed appropriately and rigorously? 

Reviewer #3: Yes

4. Have the authors made all data underlying the findings in their manuscript fully available?

Reviewer #3: Yes

5. Is the manuscript presented in an intelligible fashion and written in standard English?

Reviewer #3: Yes

6. Review Comments to the Author

Reviewer #3: The authors addressed most of my concerns by highlighting the limits of the study. Although the low number of steps still expose the results to a risk of reliability bias, the clinically based explanation of the methods is satisfactory. However, there are still some issues to be addressed before considering endorsement to publication.

The sample description is incomplete. Considering that HR has shown to be altered in subjects with H&Y ≥ 3 (Castiglia et al., 2021), providing the prevalence of disease severity subtypes is mandatory to improve the interpretability of the results. Particularly, in the discussion about the differences between your study and existing literature on HR (line 372-380), if you enrolled a higher number of subjects with H&Y < 3, it would be expected to find no improvements in HR because this parameter is usually unaltered in subjects in ON condition with low disease severity.

7. PLOS authors have the option to publish the peer review history of their article (what does this mean?). If published, this will include your full peer review and any attached files.

Reviewer #3: No

---

## [Editor Report · Acceptance letter]

4 May 2022

PONE-D-21-17999R2 

Sensitivity to gait improvement after levodopa intake in Parkinson’s disease: a comparison study among synthetic kinematic indices 

Dear Dr. Sorrentino:

I'm pleased to inform you that your manuscript has been deemed suitable for publication in PLOS ONE. Congratulations! Your manuscript is now with our production department. 

Kind regards, 

on behalf of

Dr. J. Lucas McKay 

Academic Editor

PLOS ONE